# On Task Vectors and Gradients

**Luca Zhou**[*§]    **Daniele Solombrino**[*§]    **Donato Crisostomi**[§]    **Maria Sofia Bucarelli**[§]

**Giuseppe A. D'Inverno**[†]    **Fabrizio Silvestri**[§]    **Emanuele Rodolà**[§]

luca.zhou@uniroma1.it
{solombrino, crisostomi, rodola}@di.uniroma1.it
{bucarelli, fsilvestri}@diag.uniroma1.it
gdinvern@sissa.it [*]

## Abstract

Task arithmetic has emerged as a simple yet powerful technique for model merging, enabling the combination of multiple finetuned models into one. Despite its empirical success, a clear theoretical explanation of why and when it works is lacking. This paper provides a rigorous theoretical foundation for task arithmetic by establishing a connection between task vectors and gradients of the task losses. We show that under standard gradient descent, a task vector generated from one epoch of finetuning is exactly equivalent to the negative gradient of the loss, scaled by the learning rate. For the practical multi-epoch setting, we prove that this equivalence holds approximately, with a second-order error term that we explicitly bound for feed-forward networks. Our empirical analysis across seven vision benchmarks corroborates our theory, demonstrating that the first-epoch gradient dominates the finetuning trajectory in both norm and direction. A key implication is that merging models finetuned for only a single epoch often yields performance comparable to merging fully converged models. These findings reframe task arithmetic as a form of approximate multitask learning, providing a clear rationale for its effectiveness and highlighting the critical role of early training dynamics in model merging.

## 1   Introduction

The pretrain-then-finetune paradigm has become a cornerstone of deep learning, enabling large, general-purpose models to be adapted for countless specific tasks. However, this success comes at a significant cost: storing a separate finetuned model for each task incurs substantial storage overhead, a challenge that escalates with a growing number of specialized applications. To address this, model merging has emerged as a promising solution, offering a way to combine multiple task-specific models into a single, unified model without a proportional increase in size.

Among the various merging techniques, task arithmetic [13] stands out for its elegant simplicity and surprising empirical effectiveness. This method constructs a "task vector" by taking the weight difference between a finetuned model and its pretrained base, and then builds a multitask model by summing these vectors. Despite widespread usage, a comprehensive theoretical understanding of why and when task arithmetic works has remained elusive. Prior work [33, 18, 12] has suggested that task vectors derived from shorter

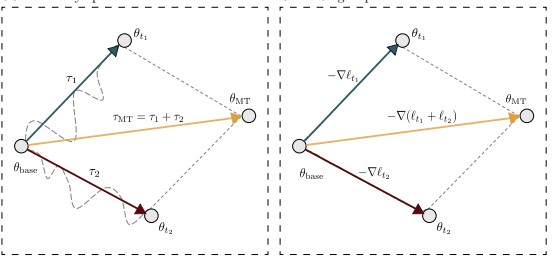

Figure 1: Left: endpoint models are finetuned with SGD for more than one epoch. Right: endpoint models are finetuned with GD for a single epoch. In this case, task vectors are equivalent to negative gradients.

---

[*]Equal Contribution, §Sapienza University of Rome, †SISSA

39th Conference on Neural Information Processing Systems (NeurIPS 2025) Workshop: UniReps.

intervals of finetunings, although less performant in a task-specific sense, are better suited for merging. However, a rigorous explanation for this phenomenon is still lacking.

In this paper, we bridge this theoretical gap by unveiling a foundational link between task vectors and the dynamics of gradient descent. Our central thesis is that task arithmetic can be understood as an approximation of simultaneous multitask learning. We establish this connection by first highlighting that, under full-batch Gradient Descent (GD), a task vector finetuned for one epoch is precisely the negative loss gradient, scaled by the learning rate. This fundamental insight implies that summing these vectors is mathematically equivalent to a single GD step on an aggregated multitask loss, where the scaling factor acts as an effective learning rate.

Building on this, we demonstrate that for multi-epoch finetuning, this equivalence still holds approximately, with a second-order deviation of $O(\eta^2)$. We provide explicit, uniform 2-norm bounds for this error term in feed-forward networks, shedding light on its structure and the factors that influence its magnitude. Our extensive empirical studies across seven vision benchmarks further validate this theory, revealing that a large portion of the gradient norm is accrued during the first epoch and that subsequent gradients remain well-aligned with this initial direction. This suggests that the early finetuning dynamics largely dictate the model's final trajectory.

A key implication of our findings is that merging models finetuned for just one epoch can achieve performance comparable to, or even better than, merging fully finetuned models. This not only offers a principled theoretical explanation for the empirical effectiveness of task arithmetic, but also provides a clear rationale for merging models finetuned for shorter intervals. Our work offers a novel perspective on model merging, re-framing it as a form of approximate multitask learning, and highlights the critical importance of early training dynamics in the finetuning process.

Our key contributions are:

- **Theoretical Foundation:** We rigorously prove that a one-epoch GD task vector is a scaled negative gradient and show that subsequent task arithmetic iterations diverge from joint multitask training iterates by only a second-order term $O(\eta^2)$.
- **Error Bounds:** We derive explicit uniform 2-norm bounds for this second-order error term in feed-forward networks, assuming bounded weights and activation functions with bounded derivatives.
- **Empirical Validation:** We empirically confirm the dominant contribution of the first-epoch gradient to the overall finetuning trajectory, both in terms of norm and direction, across a variety of vision tasks.
- **Explaining Shorter Finetuning:** We provide a theoretical rationale for why shorter finetuning periods (e.g., one epoch) are beneficial for model merging.

## 2  Task Vectors as Gradients

This section formalizes the relationship between task vectors and multitask loss gradients. For analytical clarity, we model learning as standard *full–batch* gradient descent with a fixed step size $\eta$, rather than stochastic or mini–batch variants. This assumption, common in theoretical studies of convergence and generalization [2, 21], simplifies the derivations while preserving key insights into the learning dynamics.

Vanilla task arithmetic (TA) [13] constructs a multitask model by:

(i) finetuning the base network separately on each task, and

(ii) adding the resulting task vectors, scaled by a coefficient $\alpha$, to the original pretrained weights.

The scaling parameter $\alpha$ is typically selected via hyperparameter tuning. In the full–batch setting and with a suitably scaled learning rate $\eta$, we show that TA is *exactly* equivalent to one epoch of joint training; thereafter, the two approaches diverge due to a curvature–controlled $O(\eta^2)$ term.

We first introduce the notation used throughout the rest of the paper.

**Notations.**   Let $T$ denote the set of tasks, with size $|T|$. The pretrained model weights are $\theta_{\text{base}}$. For a task $t \in T$, let $\theta_t^{(k)}$ be the parameters obtained after $k$ epochs of finetuning on $t$. The *task*

*vector* is defined as $\tau_t^{(k)} := \theta_t^{(k)} - \theta_{\text{base}}$, i.e., the parameter displacement induced by finetuning [13]. Finetuning on task $t$ minimizes the empirical loss

$$\overline{L}_t(\theta) := \frac{1}{n_t}\sum_{i=1}^{n_t}\ell(x_i, y_i, \theta),$$

where $n_t$ is the size of training data for task $t$ and $\ell$ is the per–sample loss. We denote by $\theta_{\text{MT}}^{(k)}$ the parameters obtained after $k$ epochs of *joint* training on all tasks, i.e., by minimizing the combined loss $\sum_{t\in T}\overline{L}_t(\theta)$. The corresponding *multitask vector* is $\tau_{\text{MT}}^{(k)} := \sum_{t\in T}\tau_t^{(k)}$.

We are now ready to state our theoretical results.

---

**Theorem 1.** *Let $\theta_{TA}^{(k)} = \theta_{base} + \alpha\sum_{t\in T}\tau_t^{(k)}$ be the model obtained using vanilla task arithmetics with parameter $\alpha$. Let $\{\theta_t^{(k)}\}_{t\in T}$ be produced by running $k$ full-batch GD epochs with step size $\eta$ on each task, and let $\theta_{\text{MT}}^{(k)}$ be the model obtained from $k$ GD epochs with step size $\alpha\eta$ on the aggregated loss $\sum_{t\in T}\overline{L}_t$. It holds that*

$$\theta_{TA}^{(1)} = \theta_{MT}^{(1)} \tag{1}$$

$$\theta_{TA}^{(k)} = \theta_{MT}^{(k)} + \eta^2 C(\{\theta_{MT}^{(j)}\}_{j=1}^{k-2}) + O(\eta^3) \quad \textit{for } k > 1 \tag{2}$$

*where* $C(\{\theta_{\text{MT}}^{(j)}\}_{j=1}^{h}) = \sum_{t\in T}\sum_{e=0}^{h}\nabla^2\overline{L}_t(\theta_{\text{MT}}^{(e)})\sum_{m=0}^{e}r_t(\theta_{\text{MT}}^{(m)}).$ $\tag{3}$

*with* $r_t(\theta) := \alpha\sum_{\substack{t'\neq t \\ t'\in T}}\nabla\overline{L}_{t'}(\theta) + (\alpha - 1)\nabla\overline{L}_t(\theta)$

---

Equation (1) states that, after a single epoch of gradient descent (GD), task arithmetic (TA) matches multitask training *exactly* when the step size for finetuning each task is $\eta$ and the step size for optimizing the multitask loss is $\alpha\eta$, where $\alpha$ is the scaling coefficient applied to the multitask vector when summing it to the base model in TA. Fig. 1 illustrates the intuition of the equivalence.

For later epochs ($k > 1$), the linear term in $\eta$ cancels, and the difference between the two methods reduces to a quadratic term $\eta^2 C(\cdot)$ plus higher–order corrections $O(\eta^3)$. Under our full–batch GD assumption, expanding the updates in powers of $\eta$ provides a principled, perturbative view of the dynamics. Specifically, when $\eta$ is small, the parameter difference $\theta_{TA}^{(k)} - \theta_{MT}^{(k)}$ reveals that the leading $O(\eta)$ contributions vanish, leaving a curvature–controlled $O(\eta^2)$ term together with higher–order terms. This expansion pinpoints the exact source of the TA–GD gap in the curvature term $C(\cdot)$.

## 2.1 Dominance of The First Epoch

Beyond the first epoch, even without exact equivalence to task gradients, task vectors remain effective because much of the model's finetuning trajectory is dictated by the gradient information from the first epoch. We show this in the following controlled experiment.

We compared the performance of TA obtained by merging models either at full convergence or after just one epoch of finetuning per task (Fig. 2). Remarkably, the multitask model obtained by merging models finetuned for a *single* epoch performs competitively to the one obtained by merging models finetuned to convergence across all tasks. This phenomenon can be motivated by our theoretical finding that early task vectors closely resemble the true gradients, and also suggests that *for the sake of multitask model merging, performing one epoch of finetuning is often enough, being the task vectors better approximations of the gradients.*

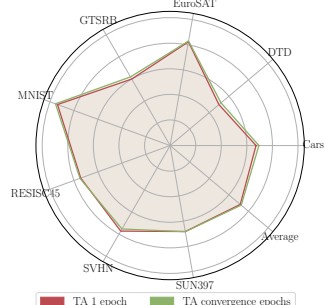

Figure 2: Task arithmetic accuracy: 1 epoch vs. converged.

Motivated by the strong performance of one-epoch merging, we next examine how much each epoch actually contributes to the overall optimization. Fig. 3a plots the epoch-wise normalized gradient norm $\frac{\|\nabla_\theta^{(k)} L\|}{\sum_{k'=1}^{K} \|\nabla_\theta^{(k')} L\|}$. We observe that the first epoch contributes the largest share of total gradient norms across training. Even in datasets where this dominance is less pronounced, such as RESISC45 and DTD, we speculate that the initial epoch still largely determines the training direction. As depicted in Fig. 3b, the gradients from the first 5 epochs maintain a high cosine similarity ($> 0.8$) with the first epoch's gradient. Thus, the effectiveness of TA arises from the alignment between fully trained and first-epoch task vectors, the latter being closer to true task-loss gradients. Fully trained vectors favor task-specific performance, while first-epoch vectors prioritize gradient approximation. Both yield similar multitask performance, but the single-epoch setting is more efficient.

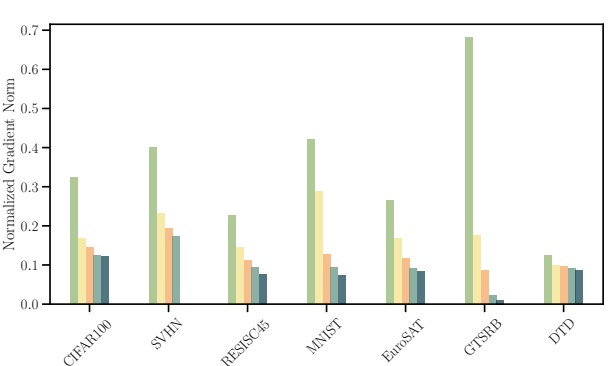

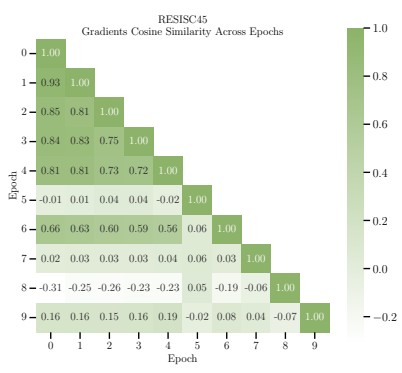

(a) Normalized gradient norms after 5 epochs of finetuning.

(b) Pairwise cosine similarities of gradients from the first 10 epochs.

Figure 3: Analysis of first-epoch gradients.

It is worth noting that our theoretical analysis does not exactly apply when more advanced optimizers (e.g. Adam) are used instead of GD. However, treating them as an approximation to GD preserves the intuition. From this view, reducing task interference corresponds to minimizing gradient conflicts in multitask learning.

## 2.2 Proof Sketch

In the following, we provide a sketch proof of Theorem 1; the detailed derivation is left to Appendix A.1. In this section we explain the main steps of the proof.

We start by presenting a key proposition that is the technical backbone for Theorem 1, which can be regarded as a corollary. We then outline the roadmap for the proof of the proposition and the theorem.

The following proposition gives a term-by-term Taylor expansion of the multitask vector.

**Proposition 1.** *Let $\{\theta_t^{(k)}\}_{t\in T}$ be produced finetuning the base model for $k$ full-batch GD epochs with step size $\eta$ for each task, and let $\theta_{\mathrm{MT}}^{(k)}$ be the model obtained from $k$ GD epochs with step size $\alpha\eta$. It holds that*

$$\tau_{MT}^{(1)} = -\eta\nabla\sum_{t\in T}\overline{L}_t(\theta_{base}) \tag{4}$$

$$\tau_{MT}^{(k)} = -\eta\sum_{t\in T}\sum_{j=0}^{k-1}\nabla\overline{L}_t(\theta_{MT}^{(j)}) + \eta^2 C(\{\theta_{MT}^{(j)}\}_{j=1}^{k-2}) + O(\eta^3) \quad \forall k \geq 2 \tag{5}$$

To better appreciate the relationship between a task vector and the gradient computed on the corresponding task dataset, consider the single-task case, where the task vector is exactly the additive inverse of the gradient, scaled by the learning rate $\eta$.

**Remark 1.** *From Proposition 1, it follows that, for a single task t, and after a single finetuning epoch, $\tau_t = -\eta \nabla \overline{L}_t(\theta_{base})$, where $\eta$ is the learning rate.*

This relationship implies that, under the theorem's assumptions, adding the task vector to the pretrained model approximates the effect of single-epoch gradient descent.

Before diving into the proof of Proposition 1 and of Theorem 1, it is helpful to provide a roadmap. Our goal is to compare the multitask training and vanilla task arithmetic, and show that they differ only by a curvature–controlled (second-order) term or higher-order terms. To improve readability, we divide the proof into the following two stages.

**Stage 1: Decomposing Single-Task Error** Vanilla task arithmetic builds its model by first finetuning each task separately, then adding the resulting task vectors $\tau_t^{(k)} = \theta_t^{(k)} - \theta_{\text{base}}$. To analyze the discrepancy between the parameters obtained by TA and MTL, it is therefore useful to understand how far the parameter point produced by the task $i$ after $k$ epochs $\theta_t^{(k)}$ lies from the multitask point of $k$-epoch $\theta_{\text{MT}}^{(k)}$. We show in Lemma 1 that this distance is shaped by (i) a linear term (the extra gradient step that task $i$ takes on its own), (ii) a quadratic curvature term, plus (iii) higher-order terms; we assume the higher-order terms can be ignored for a sufficiently small step size.

All heavy computation is done once inside Lemma 1, and expressed into the variables $r_t$, $p_t^k$, and $s_t^k$. These variables are useful to simplify the notation later and symbolize three distinct effects that appear when comparing a single-task trajectory with the multitask one. We describe them below:

- $r_t(\cdot)$ quantifies the mismatch between the two gradients at one step.

$$r_t(\theta) = \alpha \sum_{\substack{t' \neq t \\ t' \in T}} \nabla \overline{L}_{t'}(\theta) + (\alpha - 1)\nabla \overline{L}_t(\theta) = \alpha \sum_{t' \in T} \nabla \overline{L}_{t'}(\theta) - \nabla \overline{L}_t(\theta) \quad (6)$$

- $p_t^k(\cdot)$ represents the accumulation of those mismatches over many steps.

$$p_t^k(\theta_{\text{base}}, \theta_{\text{MT}}^{(1)}, \dots, \theta_{\text{MT}}^{(k)}) = \sum_{j=0}^{k} r_t(\theta_{\text{MT}}^{(j)}) \quad (7)$$

- $s_t^k(\cdot)$ accounts for the extra bending introduced by curvature once the paths have parted.

$$s_t^k(\theta_{\text{base}}, \dots, \theta_{\text{MT}}^{(k)}) = \sum_{j=0}^{k} \nabla^2 \overline{L}_t(\theta_{\text{MT}}^{(j)})[p_t^j(\theta_{\text{base}}, \dots, \theta_{\text{MT}}^{(j-1)})]. \quad (8)$$

**Lemma 1.** *Using the notation introduced in Proposition 1, it holds that*

$$\theta_t^{(1)} = \theta_{MT}^{(1)} + \eta p_t^0(\theta_{base}) \quad (9)$$

*and for $m \geq 2$:*

$$\theta_t^{(m+1)} = \theta_{MT}^{(m+1)} + \eta p_t^m(\theta_{base}, \dots, \theta_{MT}^{(m)})$$
$$- \eta^2 s_t^{m-1}(\theta_{base}, \dots, \theta_{MT}^{(m-1)}) + O(\eta^3) \quad (10)$$

The proof for $m = 1$ is straightforward, the proof for $m \geq 2$ is done by induction and it is detailed in the appendix (see Appendix A.1). We just provide here a concise and intuitive description to guide the reader clearly through the base step of the induction $m = 2$. The proof is based on expanding the single-task update around the multitask parameter using a Taylor approximation. After simplifying, we identify two distinct terms. (i) A first-order linear term ($p_t^1$), which accumulates the differences between task-specific gradients and the joint gradient from the previous epochs; (ii) A second-order curvature correction ($s_t^0$), which captures the local curvature of task $t$'s loss surface around the multitask trajectory. By rearranging these terms, we explicitly see the lemma's structure emerge for the second epoch, thus establishing the inductive base step.

**Stage** 2**: Error Aggregation across Tasks**    Once we have the difference between the parameters obtained by finetuning on single tasks and training jointly on all tasks, we are left to sum over all tasks to obtain the difference between $\theta_{TA}$ and $\theta_{MT}$. We proved that when summing over all tasks, the linear term in $\eta$ cancels out and the quadratic curvature terms sum up into the coefficient $C$ (Equation 3).

In particular, the proofs of Proposition 1 and Theorem 1 use Lemma 1 for the part relative to epoch greater than or equal to 2. Epoch 1 is handled directly: a straightforward computation shows that task arithmetic and multitask training produce the same parameter vector with a suitable choice of learning-rate scaling. Both the task arithmetic and the multitask strategies take their first update in exactly the same overall direction, so after one epoch, they land on the very same parameters. More formally,

$$\theta_{\text{TA}}^{(1)} = \theta_{\text{base}} + \alpha \sum_{t \in T} \tau_t^{(1)} = \theta_{\text{base}} - \eta\alpha \sum_{t \in T} \nabla \overline{L}_t(\theta_{\text{base}})$$

while, choosing $\alpha\eta$ as learning rate for the loss $\sum_{t \in T} \overline{L}_t$ :

$$\theta_{\text{MT}}^{(1)} = \theta_{\text{base}} - \alpha\eta \sum_{t \in T} \nabla \overline{L}_t(\theta_{\text{base}}).$$

For epoch $k \geq 2$, w plug the equations obtained in Lemma 1 into the gradient sums that define the multitask updates and then add over all tasks, the linear drift pieces cancel out, which means TA still matches the multitask iterate to first order.

The only part that survives the summation is the quadratic curvature block singled out by the lemma, and that block gives exactly the second-order gap stated in the theorem and proposition.

## 2.3   Bounding Second-Order Error Term $C(\{\theta_{\text{MT}}^{(j)}\})$

The error term $C(\{\theta_{\text{MT}}^{(j)}\}_{j=1}^{h})$ defined in Equation 3 precisely captures the second-order discrepancy between vanilla task arithmetic and true multitask training. To understand how large this discrepancy can become in practical scenarios, we now provide explicit, uniform bounds on the 2-norm of $C$. We achieve these bounds by introducing mild structural assumptions on the neural network model. Specifically, we consider feed-forward networks with bounded weights, bounded inputs, and activations with controlled first and second derivatives. Under these conditions, the error term $C$ can be explicitly bounded, as detailed in the following theorem. Very similar structural assumptions (bounded weights and inputs, Lipschitz- or controlled-derivative activations) underpin the theoretical analyses in several seminal works (E.g. [29, 3, 11]).

> **Theorem 2** (Uniform bound on the coefficient vector $C(\{\theta_{\text{MT}}^{(j)}\})$). *Assume the hypothesis of Proposition 1 holds and let $C(\{\theta_{MT}^{(j)}\}_{j=1}^{h})$ be the error term defined in Equation 3. We assume the tasks are all classification tasks optimized with cross-entropy loss. We also add a few structural constraints on the network itself. Specifically the model is a depth-L feed-forward network with weight matrices $W^{(1)}, \ldots, W^{(L)}$. For every layer, there exist positive constants $s_\ell$ such that $\|W^{(\ell)}\|_2 \leq s_\ell$. The inputs are also bounded $\|x\|_2 \leq M_x$. Finally, we require the activation functions to have bounded first and second derivatives, i. e., there are $\beta_\phi, \gamma_\phi > 0$ such that $\sup_z |\phi'(z)| \leq \beta_\phi$ and $\sup_z |\phi''(z)| \leq \gamma_\phi$. Setting $\Pi = \prod_{\ell=1}^{L} s_\ell$, we obtain the following.*
>
> *(i)* **For general activations***:*
>
> $$\left\| C(\{\theta_{\text{MT}}^{(j)}\}_{j=1}^{h}) \right\|_2 \leq T \binom{h + 2}{2} |\alpha T + 1| \, H_{\text{max}}^\phi \, G_{\text{max}}^\phi,$$
>
> *with $H_{\text{max}}^\phi \leq 2\gamma_\phi M_x^2 \Pi^2 \beta_\phi^{2L-2}$ and $G_{\text{max}}^\phi \leq \sqrt{2} M_x \Pi \beta_\phi^{L-1}$.*
>
> *(ii)* **For ReLU activations** *($\gamma_\phi = 0$, $\beta_\phi = 1$):*
>
> $$\left\| C(\{\theta_{\text{MT}}^{(j)}\}_{j=1}^{h}) \right\|_2 \leq \frac{T}{2} \binom{h + 2}{2} |\alpha T + 1| \, H_{max}^{ReLU} G_{max}^{ReLU}$$
>
> *with $H_{\text{max}}^{ReLU} \leq \frac{1}{2}\sqrt{2} M_x^3 \Pi^3 \beta_\phi^{3L-3}$ and $G_{\text{max}}^{ReLU} \leq \sqrt{2} M_x \Pi$.*

Remarkably, the bound splits into a *task-dependent* factor $T^{\frac{(h+1)(h+2)}{2}} |\alpha T - 1|$ and a *network-dependent* factor controlled by $\{M_x, \Pi, \beta_\phi, \gamma_\phi\}$. For ReLU activations, the constant improves because the network Hessian reduces to 0.

The proof, of which the complete derivation can be found in A.2, consists of the following steps:

(i) We provide a bound over the $\ell_2$ norm of $C(\{\theta_{\text{MT}}^{(j)}\}_{j=1}^h)$:

$$\|C\|_2 \leq T \frac{(h+1)(h+2)}{2} |\alpha T + 1| H_{\max}^\phi G_{\max}^\phi$$

where we define the uniform bounds:

- **Hessian bound** $H_{\max}^\phi := \max\limits_{t,\ell} \|\nabla^2 \overline{L}_t(\theta_{\text{MT}}^{(\ell)})\|_2$
- **Gradient bound** $G_{\max}^\phi := \max\limits_{t,m} \|\nabla \overline{L}_t(\theta_{\text{MT}}^{(m)})\|_2$

(ii) Then, we bound the Hessian bound $H_{\max}^\phi$ and the gradient bound $G_{\max}^\phi$ as follows:

- $H_{\max}^\phi \leq 2\,\gamma_\phi\, M_x^2\, \Pi^2 \beta_\phi^{2L-2}$ (for general activations)
- $G_{\max}^\phi \leq \sqrt{2}\, M_x\, \Pi\, \beta_\phi^{L-1}$

where we assumed that $\|x\|_2 \leq M_x$, $\|W^{(\ell)}\|_2 \leq s_\ell$, $|\phi'(z)| \leq \beta_\phi$.

(iii) Finally, plugging $H_{\max}^\phi$ and $G_{\max}^\phi$ explicitly into the inequality reproduces exactly the two cases stated in Theorem 2.

## 3 Analysis and Discussion

Having formalized the relationship between task vectors and task gradients and bounded their divergence, we now bridge this theory with empirical evidence. This section demonstrates how our framework for understanding task arithmetic as a form of approximate multitask learning provides a strong theoretical basis for several key empirical observations. We first analyze how shorter finetuning intervals lead to better mergeability, then visualize the benefits of a more controlled parameter space trajectory, and finally, we discuss why task proficiency does not necessarily correlate with mergeability. Our analysis reveals that the dynamics of early-stage training are crucial for successful model merging.

### 3.1 Empirical Advantage of Premature Task Vectors

We now discuss how the relationship between task vectors and task gradients manifests in empirical multitask performance. Our theoretical analysis suggests that task vectors obtained from shorter intervals of finetuning more closely resemble multitask gradients, making them better suited for model merging. In other words, task arithmetic applied to such "premature" task vectors can more effectively simulate the dynamics of multitask learning.

Prior work by Zhou et al. [33] provides experimental evidence consistent with this perspective. In their iterative task arithmetic framework, they start from a pretrained base model, finetune each task for a short interval to obtain task vectors, merge these vectors via task arithmetic to produce a new base model, and repeat this process over multiple rounds. They demonstrate that, under a fixed total finetuning budget, increasing the merge frequency (and thus decreasing the finetuning interval per round) consistently improves multitask performance. This aligns with our claim that early-stage task vectors are more aligned with multitask gradients and therefore yield better merged models.

The study further compares different merging methods across different budgets of finetuning, showing that iterative task arithmetic continues to enhance performance even at larger budgets, whereas other task vector-based methods quickly plateau, despite more per-task finetuning. This reinforces our perspective that the beneficial properties of early-stage task vectors, namely their closer resemblance to multitask gradients, are a key factor in successful model merging.

### 3.2 Parameter Space Trajectory

Most task arithmetic-based approaches perform aggregation in a one-shot fashion starting from the pretrained model, which can result in overshooting the multitask optimum. To better understand

the potential benefits of more controlled update trajectories, we investigate how taking smaller, gradient-similar steps influences the optimization path. This analysis further explores the effect of merging early task vectors to better approximate the true multitask gradients.

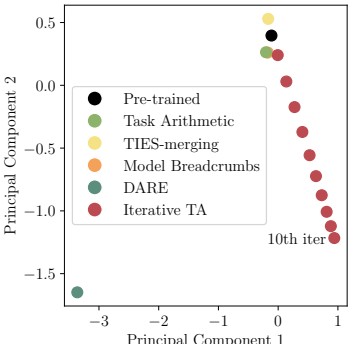

Figure 4: Checkpoint projection of different merging strategies.

Iterative task arithmetic [33], a setup where models are repeatedly merged using task vectors obtained after minimal finetuning, leads to a gradual evolution of the base model. This serves as a conceptual tool to visualize whether such updates can guide the model toward lower-loss regions. Fig. 4 shows the 2D PCA projection of merged checkpoints produced by different merging strategies. We observe that methods like TIES and Model Breadcrumbs converge to similar basins, while DARE, owing to its stochastic pruning, diverges significantly. The trajectory composed of small merging steps appears to move toward a distinct and lower-loss region.

This experiment illustrates how following directions better aligned with multitask gradients can lead to improved generalization in principle, even in the absence of shared data.

### 3.3 Task Proficiency is not Mergeability

Continuing our exploration of the parameter space trajectory, we now turn to a critical aspect: the relationship between a model's proficiency on a single task and its suitability for merging. While one might assume that more specialized, higher-performing models would lead to better merged results, our empirical findings suggest the opposite. As illustrated in Fig. 2, merging models finetuned to full convergence often yields no gains compared to merging models finetuned for just a single epoch.

This outcome aligns with the observation from the previous subsection that highly specialized models diverge further in parameter space [18], making naive aggregation less effective (see also Fig. 4). From our gradient approximation standpoint, the longer a model finetunes on its task, the more its task vector becomes a "noisy" surrogate for the true multitask gradient. In contrast, lightly finetuned models remain closer to the pretrained base, resulting in smaller, more mergeable updates. This is because, as we have shown, the first epoch's task vector is an exact scaled negative gradient, and the contributions from subsequent epochs introduce a curvature-controlled second-order error.

These findings aligns with prior work by Ortiz-Jimenez et al. [22], which attributes the effectiveness of task arithmetic to *weight disentanglement*, showing that task vectors are more reliable when models remain close to the pretrained initialization, i.e., in the tangent space, where functional components are more linearly separable. Our work offers an orthogonal perspective based on optimization dynamics rather than geometry. Both analyses converge on a similar insight: *models that stay closer to the pretrained base are more mergeable*. We attribute this advantage to task vectors being more faithful approximations of the task gradients, in contrast to their explanation based on geometric disentanglement or reduced task interference.

## 4   Related work

**Mode Connectivity and Model Merging:** Mode connectivity studies how weights characterize local minima. Frankle et al. [10] showed linear paths exist between models with shared initialization, and Entezari et al. [9] demonstrated convergence to a shared basin post-permutation. Permutation-based merging builds on this, with methods like optimal transport matching [24], `Git Re-Basin` [1], and `REPAIR` [16]. More recent approaches explore embedding-space merging [20] and cycle-consistent alignment [6]. Averaging techniques beyond simple averaging [25], such as population-based merging [15], RegMean [14], and Fisher-weighted merging [19], optimize merging coefficients in the weighted average. Post-merge performance has been shown to suffer from gradient mismatches between individual models [7]. Choshen et al. [5] introduced a method of merging finetuned models to produce an initialization superior to pretraining, leading to improved downstream performance when subsequently finetuned on a target task.

**Task Vectors:** Task vector methods [13] represent task-specific updates as deltas from a shared model, enabling arithmetic operations for transfer, forgetting, and composition. Extensions reduce

interference via sparsity [23], pruning [26, 8], masking [30], and optimization [27]. Some add test-time adaptation via learned modules [28] or advocate disentangled finetuning in tangent space [22]. While these works enhance task vectors post-hoc, several works have explored how the duration of finetuning affects model merging. Specifically, Zhou et al. [33], Lu et al. [18], Horoi et al. [12] highlighted the advantage of merging early task vectors that are finetuned for short intervals, yet exhibiting premature task-specific performance. Our work provides the first rigorous theoretical foundation for this phenomenon.

**Multitask Learning:** Multitask learning (MTL) is a learning paradigm that aims to improve the generalization performance of multiple related tasks by learning them jointly. Early foundational work [4] established that MTL can enhance performance by leveraging shared representations and inductive biases across tasks. Subsequent studies [32] have demonstrated that identifying and exploiting the structure among visual tasks can lead to significant reductions in labeled data requirements while maintaining performance. However, MTL presents optimization challenges, particularly due to gradient interference between tasks. Methods like Gradient Surgery [31] address this by orthogonally projecting gradients to mitigate conflicts. Similarly, Liu et al. [17] introduces strategies to balance objectives and ensure convergence. Our theory and experiments show task vectors closely align with task gradients, especially in single-epoch finetuning, suggesting that merging such models can effectively approximate multitask learning.

## 5 Limitations and Future Work

**Limitations:** Our analysis is grounded in the full-batch Gradient Descent (GD) setting, while in practice, task vectors are typically derived from models trained with Stochastic Gradient Descent (SGD). The inherent noise and variance introduced by SGD complicate the theoretical analysis and remain outside the scope of our current framework. Additionally, our closed-form bound on the error term is derived specifically for feed-forward networks. Although these serve as foundational models, they lack the architectural complexity and inductive biases of modern architectures such as CNNs and Transformers, limiting the direct applicability of our theoretical guarantees.

**Future Work:** Several directions emerge for extending this work. A formal treatment of task arithmetic under SGD would help bridge the gap between theory and practical training procedures. Similarly, extending our theoretical bounds to more complex architectures (CNNs, Transformers, etc.) could broaden the impact of our analysis. Further investigation into the second-order error term C, including when it becomes negligible or whether it is approximable, could improve task arithmetic. Finally, our findings suggest that early training dynamics are crucial for mergeability, implying that merging strategies might benefit from aligning with initial gradient directions. This insight connects to broader themes such as early stopping, flat vs. sharp minima, and the inductive role of pretrained models. Exploring these links may yield a more unified understanding of how optimization dynamics shape model merging.

## 6 Conclusion

In this work, we provide a rigorous theoretical foundation for task arithmetic, demystifying its empirical success by connecting task vectors directly to gradients of the task losses. We start by highlighting that a task vector generated from a single epoch of standard gradient descent is exactly equivalent to the scaled negative gradient of the task loss, and therefore, applying task arithmetic after a single epoch of per-task finetuning is equivalent to applying gradient descent in the multitask learning setting. Our central finding demonstrates that for the more practical multi-epoch scenarios, this relationship still holds approximately, with a quantifiable second-order error term that we explicitly bound for feed-forward networks. Our empirical investigation across multiple vision benchmarks corroborates this theoretical insight, revealing that the first-epoch gradient overwhelmingly dictates the finetuning trajectory in both magnitude and direction. This leads to a significant practical implication: merging models after just a single epoch of finetuning can achieve performance comparable to merging fully converged models. By reframing task arithmetic as a form of approximate multitask gradient descent, our work provides a clear rationale for its effectiveness and underscores the critical importance of early training dynamics in model merging.

**Acknowledgments**

This work is partly supported by the MUR FIS2 grant n. FIS-2023-00942 "NEXUS" (cup B53C25001030001), and by Sapienza University of Rome via the Seed of ERC grant "MINT.AI" (cup B83C25001040001).

This work is also partly supported by projects FAIR (PE0000013) and SERICS (PE00000014) under the MUR National Recovery and Resilience Plan funded by the European Union - NextGenerationEU.

G.A.D. acknowledges the support provided by the European Union - NextGenerationEU, in the framework of the iNEST - Interconnected Nord-Est Innovation Ecosystem (iNEST ECS00000043 – CUP G93C22000610007) project and its CC5 Young Researchers initiative. The views and opinions expressed are solely those of the authors and do not necessarily reflect those of the European Union, nor can the European Union be held responsible for them. In addition, G.A.D. would like to acknowledge INdAM–GNCS.

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

## A Proofs

### A.1 Proofs of Proposition 1 and Theorem 1

In this section, we provide proofs for Proposition 1 and Theorem 1. For clarity, we restate both the proposition and theorem. In the Appendix when summing the individual task losses over all tasks; sometimes we index them by their label $t \in T$, and other times we list them as $i = 1, \ldots, T$. Both notations denote the exact same aggregate over every task.

**Proposition.** *Let* $\left\{ \theta_t^{(k)} \right\}_{t=1}^{|T|}$ *be a set of models obtained by finetuning the base model* $\theta_{base}$ *for* $k$ *epochs on tasks* $T$ *using GD with a learning rate* $\eta$, *where finetuning task* $t \in T$ *minimizes the loss* $\overline{L}_t(\theta) = \frac{1}{n_t} \sum_{i=1}^{n_t} \ell(x_i, y_i, \theta)$. *Additionally, let* $\left\{ \tau_t^{(k)} \right\}_{t=1}^{|T|}$ *denote the corresponding set of task vectors, with each* $\tau_t^{(k)} = \theta_t^{(k)} - \theta_{base}$. *Let* $\tau_{MT}^{(k)}$ *be the multitask vector* $\tau_{MT}^{(k)} = \sum_{t \in T} \tau_t^{(k)}$. *Finally, let* $\theta_{MT}^{(k)}$ *represent the model obtained by minimizing the combined loss* $\sum_{i=1}^{|T|} \overline{L}_i$ *for* $k$ *epochs using GD with a learning rate of* $\alpha\eta$. *It holds that*

$$\tau_{MT}^{(1)} = -\eta \nabla \sum_{t \in T} \overline{L}_t(\theta_{base}) \tag{11}$$

$$\tau_{MT}^{(k)} = -\eta \sum_{t \in T} \sum_{j=0}^{k-1} \nabla \overline{L}_t(\theta_{MT}^{(j)}) + \eta^2 C(\{\theta_{MT}^{(j)}\}_{j=1}^{k-2}) + O(\eta^3) \tag{12}$$

*with*

$$C(\{\theta_{MT}^{(j)}\}_{j=1}^{h}) = \sum_{t \in T} \sum_{\ell=0}^{h} \nabla^2 \overline{L}_t(\theta_{MT}^{(\ell)}) \sum_{m=0}^{\ell} \left[ \alpha \sum_{t' \neq t, t' \in T} \nabla \overline{L}_t'(\theta_{MT}^{(m)}) + (\alpha - 1) \nabla \overline{L}_t(\theta_{MT}^{(m)}) \right] \tag{13}$$

**Theorem.** *Let* $\theta_{TA}^{(k)} = \theta_{base} + \alpha \sum_{t=1}^{T} \tau_t^{(k)}$ *be the model obtained using vanilla task arithmetics. Using the same notation of Theorem 1, it holds that*

$$\theta_{TA}^{(1)} = \theta_{MT}^{(1)} \tag{14}$$

$$\theta_{TA}^{(k)} = \theta_{MT}^{(k)} + \eta^2 C(\{\theta_{MT}^{(j)}\}_{j=1}^{k-2}) + O(\eta^3) \quad \text{for } k > 1 \tag{15}$$

We recall that $\theta_i^{(k)}$ is the model obtained by finetuning on task $i$ for $k$ epochs, and that both the finetuning on different tasks and the training on the average loss start from a pretrained model $\theta_{\text{base}}$.

To prove the statement of the theorem and of the corollary, we need an intermediate result. We introduce the following notation:

$$r_i(\theta, \alpha) = \alpha \sum_{j \neq i} \nabla \overline{L}_j(\theta) + (\alpha - 1) \nabla \overline{L}_i(\theta) = \alpha \sum_{j=1}^{|T|} \nabla \overline{L}_j(\theta) - \nabla \overline{L}_i(\theta) \tag{16}$$

$$p_i^k(\theta_{\text{base}}, \theta_{\text{MT}}^{(1)}, \ldots, \theta_{\text{MT}}^{(k)}) = \sum_{j=0}^{k} r_i(\theta_{\text{MT}}^{(j)}) \tag{17}$$

$$s_t^k(\theta_{\text{base}}, \ldots, \theta_{\text{MT}}^{(k)}) := \sum_{j=0}^{k} \nabla^2 \overline{L}_i\big(\theta_{MT}^{(j+1)}\big)\big[p_i^j(\theta_{\text{base}}, \theta_{\text{MT}}^{(1)}, \ldots, \theta_{\text{MT}}^{(j)})\big]. \tag{18}$$

Since $\alpha$ in this context is fixed, we will not emphasize the dependence on $\alpha$ and we will use the notation $r_i(\theta) = r_i(\theta, \alpha)$. We need the following preliminary Lemma.

**Lemma.** *Using the notation introduced in Proposition 1, it holds that*

$$\theta_i^{(1)} = \theta_{MT}^{(1)} + \eta p_i^0(\theta_{base}) \tag{19}$$

*and for $m \geq 2$*

$$\theta_i^{(m+1)} = \theta_{MT}^{(m+1)} + \eta p_i^m(\theta_{base}, \ldots, \theta_{MT}^{(m)}) - \eta^2 s_i^{m-1}(\theta_{base}, \ldots, \theta_{MT}^{(m-1)}) + O(\eta^3) \tag{20}$$

*Proof.* We first show that the statement is true for $m = 1$, and then prove the results for $m \geq 2$ by induction. In this case, the base case is given for $m = 2$. In the induction step, instead, we prove that if the statement holds for any given case $m$ then it must also hold for the next case $m + 1$.

$m = 1$. **First epoch**  For each task $i = 1, \ldots, |T|$

$$\theta_i^{(1)} = \theta_{\text{base}} - \eta \nabla \overline{L}_i(\theta_{\text{base}}) \text{ while } \theta_{\text{MT}}^{(1)} = \theta_{\text{base}} - \alpha \eta \sum_{i \in T} \nabla \overline{L}_i(\theta_{\text{base}}).$$

Consequently, it holds that

$$\theta_i^1 = \theta_{\text{MT}}^{(1)} + \eta \left[\alpha \sum_{j \neq i} \nabla \overline{L}_j(\theta_{\text{base}}) + (\alpha - 1) \nabla \overline{L}_i(\theta_{\text{base}})\right]$$

$$= \theta_{\text{MT}}^{(1)} + \eta r_i(\theta_{\text{base}}) = \theta_{\text{MT}}^{(1)} + \eta p_i^0(\theta_{\text{base}}).$$

$m = 2$. **Second epoch**

$$\theta_i^{(2)} = \theta_i^{(1)} - \eta \nabla \overline{L}_i(\theta_i^{(1)})$$

$$= \theta_{\text{MT}}^{(1)} + \eta r_i(\theta_{\text{base}}) - \eta \nabla \overline{L}_i\left(\theta_{\text{MT}}^{(1)} + \eta r_i(\theta_{\text{base}})\right)$$

$$\overset{Taylor}{\approx} \theta_{\text{MT}}^{(1)} + \eta r_i(\theta_{\text{base}}) - \eta \nabla \overline{L}_i(\theta_{\text{MT}}^{(1)}) - \frac{\eta^2}{2} \nabla^2 \overline{L}_i(\theta_{\text{MT}}^{(1)}) r_i(\theta_{\text{base}}) + O(\eta^3)$$

$$= \theta_{\text{MT}}^{(1)} - \eta \nabla \overline{L}_i(\theta_{\text{MT}}^{(1)}) + \eta r_i(\theta_{\text{base}}) - \frac{\eta^2}{2} \nabla^2 \overline{L}_i(\theta_{\text{MT}}^{(1)}) r_i(\theta_{\text{base}}) + O(\eta^3)$$

Adding and subtracting $\eta \alpha \sum_{t \in T} \nabla \overline{L}_i(\theta_{\text{MT}}^{(1)}$

$$\theta_i^{(2)} = \underbrace{\theta_{MT}^{(1)} - \eta \alpha \sum_t \nabla \overline{L}_t\big(\theta_{MT}^{(1)}\big)}_{= \theta_{MT}^{(2)}} + \underbrace{\eta \alpha \sum_{t \in T} \nabla \overline{L}_i(\theta_{\text{MT}}^{(1)}) - \eta \nabla \overline{L}_i(\theta_{\text{MT}}^{(1)})}_{\eta r_i(\theta_{MT}^{(1)})} + \eta r_i(\theta_{\text{base}})$$

$$- \frac{\eta^2}{2} \nabla^2 \overline{L}_i(\theta_{\text{MT}}^{(1)}) r_i(\theta_{\text{base}}) + O(\eta^3)$$

$$= \theta_{\text{MT}}^{(2)} + \eta r_i(\theta_{\text{MT}}^{(1)}) + \eta r_i(\theta_{\text{base}}) - \frac{\eta^2}{2} \nabla^2 \overline{L}_i(\theta_{\text{MT}}^{(1)}) r_i(\theta_{\text{base}}) + O(\eta^3)$$

$$= \theta_{\text{MT}}^2 + \eta p_i^1(\theta_{\text{base}}, \ldots, \theta_{\text{MT}}^{(1)}) - \eta^2 s_i^0(\theta_{\text{base}}) + O(\eta^3)$$

**Inductive step**  Let us assume that

$$\theta_i^{(m)} = \theta_{\text{MT}}^{(m)} + \eta p_i^{m-1}(\theta_{\text{base}}, \dots, \theta_{\text{MT}}^{(m-1)}) - \eta^2 s_i^{m-2}(\theta_{\text{base}}, \dots, \theta_{\text{MT}}^{(m-2)}) + O(\eta^3)$$

We can derive that

$$
\begin{aligned}
\theta_i^{(m+1)} &= \theta_i^{(m)} - \eta \nabla \overline{L}_i(\theta_i^{(m)}) \\
&= \theta_{\text{MT}}^{(m)} + \eta p_i^{m-1}(\theta_{\text{base}}, \dots, \theta_{\text{MT}}^{(m-1)}) - \eta^2 s_i^{m-2}(\theta_{\text{base}}, \dots, \theta_{\text{MT}}^{(m-2)}) - \eta \nabla \overline{L}_i(\theta_i^m) + O(\eta^3) \\
&= \theta_{\text{MT}}^{(m)} + \eta p_i^{m-1}(\theta_{\text{base}}, \dots, \theta_{\text{MT}}^{(m-1)}) - \eta^2 s_i^{m-2}(\theta_{\text{base}}, \dots, \theta_{\text{MT}}^{(m-2)}) \\
&\quad - \eta \nabla \overline{L}_i \left( \theta_{\text{MT}}^{(m)} + \eta p_i^{m-1}(\theta_{\text{base}}, \dots, \theta_{\text{MT}}^{(m-1)}) - \eta^2 s_i^{m-2}(\theta_{\text{base}}, \dots, \theta_{\text{MT}}^{(m-2)}) \right) + O(\eta^3) \\
&= \theta_{\text{MT}}^{(m)} + \eta p_i^{m-1}(\theta_{\text{base}}, \dots, \theta_{\text{MT}}^{(m-1)}) - \eta^2 s_i^{m-2}(\theta_{\text{base}}, \dots, \theta_{\text{MT}}^{(m-2)}) \\
&\quad - \eta \nabla \overline{L}_i(\theta_{\text{MT}}^{(m)}) - \frac{\eta}{2} \nabla^2 \overline{L}_i(\theta_{\text{MT}}^{(m)}) \left( \eta p_i^{m-1}(\theta_{\text{base}}, \dots, \theta_{\text{MT}}^{(m-1)}) - \eta^2 s_i^{m-2}(\theta_{\text{base}}, \dots, \theta_{\text{MT}}^{(m-2)}) \right) + O(\eta^3) \\
&= \theta_{\text{MT}}^{(m)} + \eta p_i^{m-1}(\theta_{\text{base}}, \dots, \theta_{\text{MT}}^{(m-1)}) - \eta^2 s_i^{m-2}(\theta_{\text{base}}, \dots, \theta_{\text{MT}}^{(m-2)}) \\
&\quad - \eta \nabla \overline{L}_i(\theta_{\text{MT}}^{(m)}) - \eta^2 \nabla^2 \overline{L}_i(\theta_{\text{MT}}^{(m)}) p_i^{m-1}(\theta_{\text{base}}, \dots, \theta_{\text{MT}}^{(m-1)}) + O(\eta^3) \\
&= \theta_{\text{MT}}^{(m+1)} + \eta p_i^m(\theta_{\text{base}}, \dots, \theta_{\text{MT}}^{(m)}) - \eta^2 s_i^{m-1}(\theta_{\text{base}}, \dots, \theta_{\text{MT}}^{(m-1)}) + O(\eta^3)
\end{aligned}
$$

$\square$

*Proof Proposition and Theorem.*  For the first epoch

$$\theta_{\text{TA}}^{(1)} = \theta_{\text{base}} + \alpha \sum_{i \in T} \tau_i^{(1)} = \theta_{\text{base}} - \eta \alpha \sum_{i \in T} \nabla \overline{L}_i(\theta_{\text{base}})$$

while, choosing $\alpha \eta$ as learning rate for the loss $\sum_{i \in T} \overline{L}_i$ :

$$\theta_{\text{MT}}^{(1)} = \theta_{\text{base}} - \alpha \eta \sum_{i \in T} \nabla \overline{L}_i(\theta_{\text{base}}).$$

So $\theta_{\text{MT}}^{(1)} = \theta_{\text{TA}}^{(1)}$.

For $k \geq 2$, notice that

$$\theta_{\text{MT}}^{(k)} = \theta_{\text{base}} - \alpha \eta \sum_{j=0}^{k-1} \nabla \sum_{t \in T} \overline{L}_i(\theta_{\text{MT}}^{(j)}). \tag{21}$$

Now, using Lemma 1, we get:

$$
\begin{aligned}
&-\alpha \eta \sum_{j=0}^{k-1} \nabla \sum_{t \in T} \overline{L}_t(\theta_{\text{MT}}^{(j)}) + \eta p_i^{k-1}(\eta^0, \dots, \eta_{\text{MT}}^{k-1}) \\
&= -\alpha \eta \sum_{j=0}^{k-1} \nabla \sum_{t \in T} \overline{L}_t(\theta_{\text{MT}}^{(j)}) + \sum_{j=0}^{k-1} r_i(\theta_{\text{MT}}^k) \\
&= -\alpha \eta \sum_{j=0}^{k-1} \nabla \sum_{t \in T} \overline{L}_i(\theta_{\text{MT}}^{(j)}) + \sum_{j=0}^{k-1} \alpha \sum_{j \in T} \nabla \overline{L}_j(\theta_{\text{MT}}^{(j)}) - \nabla \overline{L}_i(\theta_{\text{MT}}^{(j)}) \\
&= -\eta \sum_{j=0}^{k-1} \nabla \overline{L}_i(\theta_{\text{MT}}^{(j)}).
\end{aligned}
$$

Namely:

$$\theta_i^{(m+1)} = \theta_{\text{MT}}^{(m+1)} + \eta p_i^m(\theta_{\text{base}}, \ldots, \theta_{\text{MT}}^{(m)}) - \eta^2 s_i^{m-1}(\theta_{\text{base}}, \ldots, \theta_{\text{MT}}^{(m-1)}) + O(\eta^3)$$

$$= \theta_{\text{base}} - \alpha\eta \sum_{j=0}^{m} \nabla \sum_{t \in T} \overline{L}_i(\theta_{\text{MT}}^{(j)}) + \eta p_i^m(\theta_{\text{base}}, \ldots, \theta_{\text{MT}}^{(m)}) - \eta^2 s_i^{m-1}(\theta_{\text{base}}, \ldots, \theta_{\text{MT}}^{(m-1)}) + O(\eta^3)$$

$$= \theta_{\text{base}} - \eta \sum_{j=0}^{m} \nabla \overline{L}_i(\theta_{\text{MT}}^{(j)}) - \eta^2 s_i^{m-1}(\theta_{\text{base}}, \ldots, \theta_{\text{MT}}^{(m-1)}) + O(\eta^3)$$

we can rewrite the tasks vectors as

$$\tau_i^{(k)} = \theta_i^{(k)} - \theta_{\text{base}} \tag{22}$$

$$= -\eta \sum_{j=0}^{k-1} \nabla \overline{L}_i(\theta_{\text{MT}}^{(j)}) - \eta^2 s_i^{k-2}(\theta_{\text{base}}, \ldots, \theta_{\text{MT}}^{(k-2)}) + O(\eta^3) \tag{23}$$

Consequently the model obtained with TA is

$$\theta_{\text{TA}}^{(k)} = \theta_{\text{base}} + \alpha \sum_{i \in T} \tau_i^{(k)}$$

$$= \theta_{\text{base}} - \eta\alpha \sum_{j=0}^{k-1} \sum_{i \in T} \nabla \overline{L}_i(\theta_{\text{MT}}^{(j)}) - \alpha \sum_{i \in T} \eta^2 s_i^{k-2}(\theta_{\text{base}}, \ldots, \theta_{\text{MT}}^{(k-2)}) + O(\eta^3)$$

$$= \theta_{\text{MT}}^{(k)} - \alpha \sum_{i \in T} \eta^2 s_i^{k-2}(\theta_{\text{base}}, \ldots, \theta_{\text{MT}}^{(k-2)}) + O(\eta^3) \,.$$

$\square$

## A.2 Proofs of Theorem 2

In this section, we provide the proof for Theorem 2, which is restated for clarity.

**Theorem** (Uniform bound on the coefficient $C$). *Let $C(\{\theta_{MT}^{(j)}\}_{j=1}^h)$ be the error term obtained in Theorem 1, with the condition of the theorem above. Additionally, we add that the tasks are all classification tasks for which we used cross entropy loss. We also assume that the network is a depth-$L$ feed-forward network with weight matrices $W^{(1)}, \ldots, W^{(L)}$, with the property that there exist constants $s_\ell > 0$ such that $\|W^{(\ell)}\|_2 \leq s_\ell$. We abbreviate $\Pi := \prod_{\ell=1}^{L} s_\ell$. We also assume that every input vector satisfies $\|x\|_2 \leq M_x$. We assume that the activation functions have bounded first and second derivatives: there are $\beta_\phi, \gamma_\phi > 0$ such that $\sup_z |\phi'(z)| \leq \beta_\phi$ and $\sup_z |\phi''(z)| \leq \gamma_\phi$. For the ReLU activation we have $\gamma_\phi = 0$.*

   *(i) General activations.*

$$\left\| C(\{\theta_{\text{MT}}^{(j)}\}_{j=1}^h) \right\|_2 \leq T \, \frac{(h+1)(h+2)}{2} \, |\alpha T + 1| \, H_{\max} \, G_{\max},$$

   *with*

$$G_{\max} \leq \sqrt{2} \, M_x \, \Pi \, \beta_\phi^{L-1}, \qquad H_{\max} \leq 2 \, \gamma_\phi \, M_x^2 \, \Pi^2 \, \beta_\phi^{2L-2}.$$

   *(ii) ReLU activations ($\gamma_\phi = 0$).*

$$\left\| C(\{\theta_{\text{MT}}^{(j)}\}_{j=1}^h) \right\|_2 \leq T \, \frac{(h+1)(h+2)}{2} \, |\alpha T + 1| \, \frac{1}{2} \sqrt{2} \, M_x^3 \, \Pi^3 \, \beta_\phi^{3L-3}.$$

*Proof.* We want to bound the 2-norm of the coefficient $C$ in Equation 3. Let's start by noticing that

$$\|C(\{\theta_{MT^{(j)}}\}_{j=1}^h)\|_2 \leq \left\| \sum_{t \in T} \sum_{\ell=0}^{h} \nabla^2 \overline{L}_t(\theta_{\text{MT}}^{(\ell)}) \right\|_2 \cdot \left\| \sum_{m=0}^{\ell} \left[ \sum_{t' \neq t, t' \in T} \alpha \nabla \overline{L}_{t'}(\theta_{\text{MT}}^{(m)}) + (\alpha - 1) \nabla \overline{L}_t(\theta_{\text{MT}}^{(m)}) \right] \right\|_2$$

$$\leq \Big(\sum_{t\in T}\sum_{\ell=0}^{h}\|\nabla^2 \overline{L}_t(\theta_{\text{MT}}^{(\ell)})\|_2\Big)\cdot\Big\|\sum_{m=0}^{\ell}\sum_{t'\in T}\alpha\nabla\overline{L}_{t'}(\theta_{\text{MT}}^{(m)})-1\nabla\overline{L}_t(\theta_{\text{MT}}^{(m)})\Big\|_2$$

$$\leq \Big(\sum_{t\in T}\sum_{\ell=0}^{h}\|\nabla^2 \overline{L}_t(\theta_{\text{MT}}^{(\ell)})\|_2\Big)\cdot\Big\|\sum_{m=0}^{\ell}\sum_{t'\in T}\alpha\nabla\overline{L}_{t'}(\theta_{\text{MT}}^{(m)})-1\nabla\overline{L}_t(\theta_{\text{MT}}^{(m)})\Big\|_2$$

Let us denote by $G_{max}$ the max of the gradient and by $H_{max}$ the max of the Hessian, the term $C$ can be upperbounded by

$$\|C(\{\theta_{MT^{(j)}}\}_{j=1}^{h})\|_2 \leq \sum_{t\in T}\sum_{\ell=0}^{h}H_{max}\sum_{m=0}^{\ell}|\alpha T+1|G_{max} \tag{24}$$

$$= T\frac{(h+1)(h+2)}{2}(\alpha T+1)H_{max}G_{max} \tag{25}$$

We are left with bounding $H_{max}$ and $G_{max}$.

Let us start with $G_{max}$, the cross entropy loss on a sample $(x,y)$ is $L(x,y,\theta) = -\log(\sigma(f_\theta(x)\cdot e_y))$, with $\sigma(\cdot)$ being the softmax function and $e_y$ being the one-hot vector of class $y$. We denote $z = f_\theta(x)$, we denote by

$$p = \sigma(z) = \left[\frac{e^{-z_1}}{\sum_{k=1}^{K}e^{-z_k}},\dots,\frac{e^{-z_K}}{\sum_{k=1}^{K}e^{-z_K}}\right].$$

The gradient of the loss wit respect to the output of the network $z$ is:

$$g = \nabla_z(-\log(\sigma(z)\cdot e_y)) = p - e_y.$$

$$H_{logits} = \nabla_z^2(-\log(\sigma(z)\cdot e_y)) = \text{diag}(p) - pp^T$$

$K$ is the number of classes and $P$ the number of parameters. Moreover we denote by $J_\theta f_\theta(x) \in \mathbb{R}^{K\times P}$ the Jacobian of the network with respect to the parameters, $\nabla_\theta^2 f_\theta$ is a 3-dimensional tensor in $\mathbb{R}^{K\times P\times P}$. By chain rule we can obtain that:

$$\nabla_\theta L(x,y,\theta) = \underbrace{\nabla_z L(x,y,\theta)}_{g^T}\underbrace{\nabla_\theta z}_{J_\theta f_\theta(x)} = g^T J_\theta f_\theta(x) \tag{26}$$

$$\nabla_\theta^2 L(x,y,\theta) = \nabla_\theta g^T J_\theta f_\theta(x) + g^T \nabla_\theta^2 f_\theta(x) \tag{27}$$

$$= [H_{logits}J_\theta f_\theta(x)]^T J_\theta f_\theta(x) + g^T \nabla_\theta^2 f_\theta(x) \tag{28}$$

$$= J_\theta f_\theta(x)^T H_{logits}J_\theta f_\theta(x) + g^T \nabla_\theta^2 f_\theta(x) \tag{29}$$

**Bound on the Jacobian of the network** We assume the activation has bounded first and second derivative, more formally we assume the activation function $\phi(\cdot)$ is such that there exists $\beta_\phi$ so that $\sup_{x\in\mathbb{R}}\phi'(x)\leq\beta_\phi$ and there exists $\gamma_\phi$ so that $\sup_{x\in\mathbb{R}}\phi''(x)\leq\gamma_\phi$. Notice that this covers both the case of ReLU, the hyperbolic tangent or sigmoid activation function.

Indeed

for the points $x\in\mathbb{R}$ where the derivative are defined: $|\text{ReLU}'(x)|\leq 1$ and $\text{ReLU}''(x) = 0$. (30)

For the sigmoid activation function

$$\sup_{x\in\mathbb{R}}|\sigma'(x)|\leq\frac{1}{4}\text{ and }\sup_{x\in\mathbb{R}}|\sigma''(x)|\leq\frac{1}{6\sqrt{3}}. \tag{31}$$

Finally for the hyperbolic tangent

$$\sup_{x\in\mathbb{R}}|\tanh'(x)|\leq 1\text{ and }\sup_{x\in\mathbb{R}}|\tanh''(x)|\leq\frac{4}{3\sqrt{3}}. \tag{32}$$

Under the hypothesis of activation function with bounded first and second derivatives almost everywhere, plus the hypothesis that the input of the network lies in a bounded set, $\|x\|\leq M_x$, and that for each layer $\ell$ there exists $s_\ell$ so that $\|W^\ell\|_2\leq s_\ell$. We obtain, by the chain rule:

$$\|J_\theta f_\theta(x)\|\leq M_x\prod_{l=1}^{L}s_l\beta_\phi^{L-1}.$$

We denote by $\Pi = \prod_{l=1}^{L} s_l$. From this we can derive the following **bound on the gradient of the loss function**

$$||\nabla_\theta L||_2 \leq ||g||_2 ||J_\theta f_\theta(x)||_2 \leq \sqrt{2} M_x \Pi \beta_\phi^{L-1}$$

**Bound on the Hessian of the loss.**

If the activation function is the ReLU function then the network is piecewise linear and $\nabla_\theta^2 f_\theta(x) = 0$ is almost everywhere. So the Hessian of the loss becames:

$$\nabla_\theta^2 L(x, y, \theta) = J_\theta f_\theta(x)^T H_{logits} J_\theta f_\theta(x).$$

The matrix $H_{logits}$ is the correct covariance of a categorical variable, it is positive semi-definitive, so it is diagonalizable with non-negative eigenvalues. For any unit vector $x \in \mathbb{R}^n$,

$$x^\mathsf{T} H_{logits} x = \sum_{i=1}^{n} p_i x_i^2 - \left( \sum_{i=1}^{n} p_i x_i \right)^2.$$

We can look for the eigenvector corresponding to the maximum eigenvalues in the space orthogonal to the null-space. So, the maximum eigenvalues is given by:

$$\max_{x:||x||_2=1} x^\mathsf{T} H_{logits} x = \max_{x:||x||_2=1} \sum_{i=1}^{n} p_i x_i^2 - \left( \sum_{i=1}^{n} p_i x_i \right)^2. \tag{33}$$

Now, for $\|x\|_2 = 1$, we can derive that

$$\sum_{i=1}^{n} p_i x_i^2 - \left( \sum_{i=1}^{n} p_i x_i \right)^2 = \sum_{i=1}^{n} p_i x_i^2 - \sum_{i,j} p_i p_j x_i x_j = \sum_{i=1}^{n} p_i x_i^2 - \sum_i p_i^2 x_i^2 - \sum_{i \neq j} p_i p_j x_i x_j$$

$$= \sum_{i=1}^{n} p_i(1 - p_i) x_i^2 - \sum_{i \neq j} p_i p_j x_i x_j \leq \sum_{i=1}^{n} p_i(1 - p_i) x_i^2 + \frac{1}{2} \sum_{i \neq j} p_i p_j (x_i^2 + x_j^2)$$

$$= \sum_{i=1}^{n} p_i(1 - p_i) x_i^2 + \sum_{i \neq j} p_i p_j x_i^2 = \sum_{i=1}^{n} p_i(1 - p_i) x_i^2 + \sum_i p_i(1 - p_i) x_i^2$$

$$= 2 \sum_{i=1}^{n} p_i(1 - p_i) x_i^2 \leq 2 \frac{1}{4} \sum_{i=1}^{n} x_i^2 = \frac{1}{2}$$

So the maximum eigenvalue for $H_{logits}$ is $\frac{1}{2}$. Consequently, when the activation function is ReLU:

$$||\nabla_\theta^2 L(x, y, \theta)||_2 \leq \frac{1}{2} M_x^2 \Pi^2 \beta_\phi^{2L-2}.$$

In the other cases we also have to include the terms coming from the hessian of the netwrok with respect to the parameters. We have to bound that term.

**Bound Hessian of the Netwrok**    We start by noticing the structure of the Jacobian blocks. We use the following layer-wise notation:

$$h^{(\ell)} := W^{(\ell)} a^{(\ell-1)}, \qquad a^{(\ell)} := \phi(h^{(\ell)}), \qquad a^{(0)} := x.$$

Let $\Theta$ collect *all* scalar parameters and denote a single one by $\theta_p$. We consider $\theta_p^r$ is an entry of $W^{(r)}$ with $1 \leq r \leq k$.

The product splits at layer $r$:

$$\partial_{\theta_p^r} a^{(k)} = \left( D^{(k)} W^{(k)} \cdots D^{(r)} \right) \left( \partial_{\theta_p} W^{(r)} \right) a^{(r-1)}$$

where $D^{(\ell)} = \text{diag}(\phi'(h^{(\ell)}))$.

For any parameter pair $(\theta_p^r, \theta_q^{r'})$ with $1 \le r \le r' \le k$ (the function implemented by the neural network is continuous, therefore the mixed double derivatives are equal) :

$$\partial_{\theta_q^{r'}} \partial_{\theta_p^r} a^{(k)} = \underbrace{\partial_{\theta_q^{r'}} \left( D^{(k)} W^{(k)} \cdots D^{(r)} \right) (\partial_{\theta_p^r} W^{(r)}) a^{(r-1)}}_{(A)} +$$

$$\underbrace{D^{(k)} W^{(k)} \cdots D^{(r)} (\partial_{\theta_p^r} W^{(r)}) \, \partial_{\theta_q^{r'}} a^{(r-1)}}_{=0} +$$

$$\underbrace{D^{(k)} W^{(k)} \cdots D^{(r)} \, \partial_{\theta_q^{r'}} (\partial_{\theta_p^r} W^{(r)}) \, a^{(r-1)}}_{(B)}.$$

$$\partial_{\theta_q^{r'}} \partial_{\theta_p^r} a^{(k)} = \underbrace{\left( \partial_{\theta_q^{r'}} Z_{k \leftarrow r} \right) \left( \partial_{\theta_p^r} W^{(r)} \right) a^{(r-1)}}_{(A)} +$$

$$\underbrace{Z_{k \leftarrow r} \, \partial_{\theta_q^{r'}} \left( \partial_{\theta_p^r} W^{(r)} \right) a^{(r-1)}}_{(B)},$$

where

$$Z_{k \leftarrow r} := D^{(k)} W^{(k)} \cdots D^{(r)}, \qquad D^{(\ell)} = \mathrm{diag}\big( \phi'(h^{(\ell)}) \big).$$

We recall that

$$\|D^{(\ell)}\|_2 \le \beta, \qquad \|\partial_\theta W^{(\ell)}\|_2 = 1, \qquad \|W^{(\ell)}\|_2 \le s_\ell, \qquad \|a^{(r-1)}\|_2 \le M_x.$$

Define $\Pi = \prod_{\ell=1}^{k} s_\ell, \quad \Pi_{1:r-1} := \prod_{\ell=1}^{r-1} s_\ell, \quad \Pi_{r+1:k} := \prod_{\ell=r+1}^{k} s_\ell.$

**Bound of the term (A)**

$$\partial_{\theta_q^{r'}} Z_{k \leftarrow r} = \sum_{\ell=r'}^{k} \left[ D^{(k)} W^{(k)} \cdots \left( \partial_{\theta_q} D^{(\ell)} \right) \cdots D^{(r)} + D^{(k)} W^{(k)} \cdots \left( \partial_{\theta_q^{r'}} W^{(\ell)} \right) \cdots D^{(r)} \right] =$$

$$= D^{(k)} W^{(k)} \cdots \left( \partial_{\theta_q^{r'}} W^{(r')} \right) \cdots D^{(r)} + \sum_{\ell=r'+1}^{k} D^{(k)} W^{(k)} \cdots \left( \partial_{\theta_q} D^{(\ell)} \right) \cdots D^{(r)},$$

with

$$\partial_{\theta_q} D^{(\ell)} = \mathrm{diag}\big( \phi''(h^{(\ell)}) \big) \partial_{\theta_q} h^{(\ell)} =$$

$$= \mathrm{diag}\big( \phi''(h^{(\ell)}) \big) W^{(\ell)} D^{(\ell-1)} \cdots D^{(r')} \partial_{\theta_q^{r'}} W^{(r')} a^{(r'-1)}$$

Hence

$$\left\| \partial_{\theta_q^{r'}} D^{(\ell)} \right\|_2 \le \gamma \, \beta^{\ell - r'} M_x \, \Pi_{r'+1:\ell},$$

and therefore

$$\|(A)\|_2 \le M_x \Big[ \beta^{k-r+1} \Pi_{r:r'} \Pi_{r'+1:k} + \sum_{\ell=r'+1}^{k} \gamma \beta^{\ell-r'} M_x \Pi_{r'+1:\ell} \Big].$$

**Bound of the term (B)**

$$\partial_{\theta_q^{r'}} \big( \partial_{\theta_p^r} W^{(r)} \big) = 0$$

**Bound Hessian**

$$\left\| \partial_{\theta_q^{r'}} \partial_{\theta_p^r} a^{(k)} \right\|_2 \le M_x \Big[ \beta^{k-r+1} \Pi_{r:r'} \Pi_{r'+1:k} + \sum_{\ell=r'+1}^{k} \gamma \beta^{\ell-r'} M_x \Pi_{r'+1:\ell} \Big]$$

$\square$

