# OpenReview forum: "On Task Vectors and Gradients"
_NeurIPS.cc/2025/Workshop/UniReps — UniReps2025_

### Official Review · Reviewer_iFxT · 2025-09-16
**An Intuitive Theory with Useful Empirical Connections**

**Confidence:** 4

**Review:**

# Summary

Introduces two theorems:
1. Shows that task vectors [1] approximate fine-tuning gradients, and the approximation is better for lower epochs of fine-tuning.
2. Derives upper bounds for the difference between task vectors and the true fine-tuning gradient.

Then goes on to cite empirical evidence of this gradient approximation and uses it to argue that less fine-tuning yields better merging.

Overall, I think this provides a useful perspective with nice bridges from theory to application.

# Strengths

- An intuitive theory with very clear explanations. Thank you for the excellent writing.
- Very useful connections with empirical results, both corroborating the theory and providing practical insights.

# Weaknesses

- Although you prove an upper bound on error term C, you don't give any insight into what are the dominant terms in this bound or what are the implications of this bound. I guess the main takeaway is just that a bound is possible? I'm not sure how much this strengthens the conclusions of Section 3, over and above Theorem 1 itself.
- More details for reproducing the experiments would be appreciated.
- Figure 4 should be associated with the models' loss to be more convincing.

# Questions / Comments

- Section 2.1: You don't provide enough details to reproduce these experiments. Would they be reproducible from your code repo? Is it possible to provide instructions there, or in the Appendix?
- Section 2.3: You say, "For ReLU activations, the constant improves because the network Hessian reduces to 0." Do you mean the activation function Hessian?
- Fig. 3a: Could use a legend or caption saying which colors are Epochs {1, 2, 3, 4, 5}.
    - Also, Figure 3 label is missing.
- Fig. 4 seems intriguing, but you do not show the loss associated with each of these points, so we don't know that Iterative TA is actually guiding the model toward lower-loss regions.

[1] Ilharco et al. Editing models with task arithmetic. ICLR 2022.

**Score:**

4

**Topic Fit:**

3

---

### Official Review · Reviewer_FTZd · 2025-09-16
**Review of Submission9 by Reviewer FTZd**

**Confidence:** 3

**Review:**

**Summary:** This paper develops a theoretical explanation for how task arithmetic—merging models after a single epoch or multiple iterations of gradient descent—operates compared to joint training on all tasks, providing justification for certain empirical findings in the literature.

**Strengths:** The paper provides a detailed and easy-to-follow theoretical derivation of the main claims, showing that a task vector generated from one epoch of fine-tuning is equivalent to the negative gradient of the loss, scaled by the learning rate. For multi-epoch setting, it is proven that this equivalence holds approximately, with a second-order error term bounded for feed-forward networks (although not extended to practical settings where typically LLMs and Transformers are employed). The presented experimental results (somewhat limited) support the theoretical ones. The supplementary material includes further details of the proofs, and code is provided, facilitating the reproducibility of the experiments.

**Weaknesses:** The analysis of task arithmetic under full-batch gradient descent and certain assumptions allows for simplified, tractable derivations, but provides limited practical relevance and insight compared to works such as [1] and [2] already considering settings with LLMs/Transformers. The discussion of the related work may be improved by providing a more detailed contrast with the already referenced [1] for example, as both works connect task arithmetic and fine-tuning in a similar way.
Although the theoretical results as main contributions are described with clarity, the presentation of the experimental results could be improved. Specifically, it would be helpful to clarify the network architectures and the activation functions used in the experiments (which could be included in the supplementary material). For consistency, it would help to use the same datasets across Figures 2 and 3 (e.g., including CIFAR-100 in Figure 2), and to indicate what the five separate bars for each dataset in Figure 2 represent.
Finally, the supplementary material contains a few typos (e.g., in rows 516, 518, and 538, also in the equation after line 477) that should be corrected.

**Question:** Theorem 2 assumes activation functions with bounded first and second derivatives, which strictly speaking would exclude the standard ReLU because of its non-differentiability at 0. However, a ReLU-specific bound is mentioned. Could the authors detail whether this non-differentiability or higher-order terms affect the validity of that bound, and if so, in what way?

[1] Model merging by uncertainty-based gradient matching, https://arxiv.org/abs/2310.12808

[2] When is task vector provably effective for model editing? A generalization analysis of nonlinear transformers, https://arxiv.org/abs/2504.10957

**Score:**

3

**Topic Fit:**

3

---

### Official Review · Reviewer_DbiM · 2025-09-18
**Review of this paper**

**Confidence:** 4

**Review:**

**Summary**

The paper formalizes task arithmetic: after one epoch of full-batch GD, the task vector equals a negative gradient step; for multiple epochs, the gap is second-order with an explicit bound. Results on vision tasks suggest early-epoch vectors merge well.

**Strengths**

* Clear one-epoch equivalence; helpful reframing of merging as approximate multi-task GD.
* Provides explicit curvature term $C(\cdot)$ with assumptions/bounds for feed-forward nets.

**Main correctness issues**

1. **Spurious $1/2$ in the expansion of $\nabla L$.** Eq. (5)/(12)/(15) carry $\eta^2/2$ from a first-order Taylor of the gradient, which should not have $1/2$. Remove the factor and update all downstream $\eta^2$ constants.
2. **ReLU Hessian drops a necessary term.** The paper concludes $\nabla^2_\theta L=J^\top H_{\text{logits}}J$ for ReLU by asserting $\nabla^2_\theta f=0$ a.e.; but earlier the full identity includes the extra $g^\top\nabla^2_\theta f$ term, which does not vanish w\.r.t. parameters. Either bound this term or label the result Gauss–Newton.
3. **Index mismatch in Eq. (12).** Outer sum is over tasks $t$ but the inner gradient uses $i$; should be $\nabla \bar L_{t}(\cdot)$.
4. **Extra $|T|$ in $r_i(\theta,\alpha)$.** The second equality of (16) erroneously inserts $|T|$. Use $\alpha\sum_j\nabla L_j(\theta)-\nabla L_i(\theta)$.
5. **Wrong evaluation point in $p_i^k$.** (17) sums $r_i^{(k)}(\theta_{MT})$ instead of $r_i(\theta^{(j)}_{MT})$.

**Score:**

3

**Topic Fit:**

2